# Delamination of Plasticized Devices in Dynamic Service Environments

**DOI:** 10.3390/mi15030376

**Published:** 2024-03-11

**Authors:** Wenchao Tian, Xuyang Chen, Guoguang Zhang, Yuanming Chen, Jijun Luo

**Affiliations:** 1Key Laboratory of Electronic Equipment Structure Design (MOE), School of Mechano-Electronic Engineering, Xidian University, Xi’an 710071, China; 2Guangzhou Research Institute, Xidian University, Guangzhou 510000, China; 22041212749@stu.xidian.edu.cn; 3Foshan Blue Rocket Electronics Co., Ltd., Foshan 528000, China; zhangguoguang@fsbrec.com (G.Z.); luojijun@fsbrec.com (J.L.); 4Sharetek Industrial Equipment Co., Ltd., Shanghai 201109, China; frankchen@sharetek.com.cn

**Keywords:** delamination, moisture, thermal stress, RDL, fillers, block copolymer, conjugated polymer

## Abstract

With the continuous development of advanced packaging technology in heterogeneous semiconductor integration, the delamination failure problem in a dynamic service environment has gradually become a key factor limiting the reliability of packaging devices. In this paper, the delamination failure mechanism of polymer-based packaging devices is clarified by summarizing the relevant literature and the latest research solutions are proposed. The results show that, at the microscopic scale, thermal stress and moisture damage are still the two main mechanisms of two-phase interface failure of encapsulation devices. Additionally, the application of emerging technologies such as RDL structure modification and self-healing polymers can significantly improve the thermal stress state of encapsulation devices and enhance their moisture resistance, which can improve the anti-delamination reliability of polymer-based encapsulation devices. In addition, this paper provides theoretical support for subsequent research and optimization of polymer-based packages by summarizing the microscopic failure mechanism of delamination at the two-phase interface and introducing the latest solutions.

## 1. Introduction

Since the Moore Law was amended in 1975 [1], the semiconductor IC industry has doubled chip integration complexity every two years by continually miniaturizing COMS device transistor gate sizes. By 2023 the international chip foundry giant TSMC (Tsmc) process has reached 3 nm node, but its foundry Apple Inc (Apple Inc) A17 chip yield is only 55%, and its performance of the device reliability problem is far greater than the performance of the chip process to improve the climb advantage. To address this dilemma, the ITRS (International Technology Road-map for Semiconductors) proposed “More than Moore” as early as 2005 [2], that is, the use of diverse packaging forms to integrate multiple chips in a system to realize system-in-package (SIP). This development trend has gradually evolved into various advanced packaging technologies represented by Heterogeneous Integration Packaging (HIP). However, from the simplest plastic package to advanced packages under heterogeneous integration, the delamination failure phenomenon caused by the interface between the combination of two packaging materials in the dynamic service environment (i.e., temperature, stress, strain size, and distribution of the working environment over time) has become increasingly serious [3,4,5], This is gradually becoming one of the major challenges affecting the reliability of packaged devices.

Delamination failure is a low-cycle ratchet fatigue failure of the different material layers of the encapsulated device under various severe working conditions, such as high temperature, temperature cycling, and humidity, which is manifested by repeated thermal expansion and contraction, and hot and humid stress attacks [6,7]. Researchers’ studies on delamination mechanisms have mainly focused on thermal stress damage and moisture damage. Explaining the delamination phenomenon from the stress–strain perspective alone, it has been found that, at the delamination interface, mainly due to the interface between the two phases of the material, the plastic deformation of the two-phase material is formed under the action of the alternating shear stress, which ultimately leads to the failure of delamination [8,9,10,11,12,13,14]. At the same time, under the intrusion of moisture, ionic contaminants corrode the chip, and moisture thermal expansion is also the main reason for the device to appear to suffer from the “popcorn” effect [15,16,17,18,19]. However, with the continuous development of advanced encapsulation technology, the delamination mechanism of encapsulated devices should be more than limited to the macro-scale, and the micro-delamination mechanism also needs to be gradually improved, with a view to establishing a perfect delamination prevention, identification, and improvement mechanism.

## 2. Delamination Mechanism at the Two-Phase Interface

Delamination of a molded device is often considered a service life failure, i.e., a failure that occurs only during normal use of the device. A typical example is during reflow soldering, where water vapor generated at high temperatures penetrates into the interior of the plastisol, resulting in localized swelling. Delving deeper into the delamination phenomenon, we can find that its essence is due to the increase in internal stresses, resulting in the initial cracks generated in the plastisol defects developing into crack expansion. The delamination phenomenon on the surface can be categorized into three main types: (1) EMC/Si chip delamination; (2) polymer/Cu substrate delamination; and (3) Si chip/conductive adhesive delamination (Figure 1). However, to understand the generation mechanism of such delamination failures, we need to consider two main scenarios: (1) moisture damage, i.e., delamination caused by the intrusion of moisture; and (2) thermal stress damage, i.e., delamination triggered by thermal stresses due to temperature changes.

### 2.1. Damage by Moisture

Organic polymeric materials, such as resins, are widely used in a variety of encapsulation techniques, primarily to mitigate stresses due to thermal strains generated at the interface of two phases. However, this also leads to the non-gas-tight nature of electronic packages, especially due to the penetration of moisture. The gradual penetration of moisture occurs in two main cases: (1) the gradual penetration of moisture along the two-phase interface between the plasticized encapsulation material and other materials; and (2) the entry of tiny voids formed by resin polymer bonds into the interior of the device (as shown in Figure 2). These tiny voids are by-products of the stable cross-linked network structure formed during the polymer cross-linking reaction due to bonding between linear molecules. These cavities exhibit excellent hygroscopic properties. They can gradually absorb moisture in hot and humid working environments until they reach saturation.

Because the large number of molecular scale voids inside the polymer material can account for 2.5% of the entire volume of the polymer, some non-polar gases such as O_2_ can travel smoothly in the voids to reach the interior. However, polar molecules such as H_2_O will form hydrogen bonds with the hydroxyl group, which is a polar functional group inside the polymer, resulting in wet swelling and decomposition. It has been found that the water morphology in the polymer as shown in Figure 2 mainly consists of three types: (1) hydrogen bonding with the functional groups on the surface of the micropores to form bonded water; (2) adsorption of adsorbed water with the polymer molecular chain due to the capillary effect; and (3) free water inside the micropores [22,23] Using the Vander Waals volume, Cai estimated that the density of water in the special state in the plastic sealer is 100 times the density of water vapor in the standard state, which is 8% of the density of liquid water [24]. This indicates that once the packaging device saturated moisture absorption in the reflow soldering 200 ~ 250 ℃ at high temperatures, in which the combined water into water vapor, "popcorn" effect is inevitable.

#### 2.1.1. How to Characterize Moisture Diffusion

In the process of moisture diffusion, it is difficult to obtain the exact value of the saturated moisture concentration and humidity diffusion coefficient of the material through the experimental method, which is limited to the local temperature and humidity, as well as the pressure of the experiment. Nowadays, the mainstream characterization method is to use CAE (computer-aided engineering) in FEA (finite element analysis) software (such as ANSYS 2022R2, ABAQUS, etc.) to simulate the moisture diffusion process.

Researchers initially applied Fick’s second law to derive a wet diffusion model by analogy with thermal diffusion:(1)$$\frac{\partial C}{\partial t}=D\left(\frac{\partial^{2}C}{\partial x^{2}}+\frac{\partial^{2}C}{\partial y^{2}}+\frac{\partial^{2}C}{\partial z^{2}}\right)$$
where C(x,y,z) is the moisture concentration inside the material, D is the diffusion constant, and (x,y,z) is the coordinate of a point in the internal space of the material. For single polymer-based materials, bottom fillers are generally a good fit for Fick diffusion in terms of moisture absorption; however, for mold compounds, no Fick diffusion is evident, even during standard testing. In electronic packaging, the discontinuity of materials under heterogeneous integration also causes the discontinuity of moisture concentration diffusion, and Fick’s law cannot be applied simply by analogy with thermal diffusion [25,26]. So Wong et al. [27] proposed the normalization method: introduce the normalized wet concentration W (the ratio of wet concentration to saturated wet concentration, $W=C/C_{sat}$) to solve the problem of discontinuity of different interfacial parameters of the two phases of the material:(2)$$\frac{\partial W}{\partial t}=D\left(\frac{\partial^{2}W}{\partial x^{2}}+\frac{\partial^{2}W}{\partial y^{2}}+\frac{\partial^{2}W}{\partial z^{2}}\right)$$

He [28] used Fick’s second law to characterize moisture diffusion at lower temperatures when experimentally expressing the hygroscopic thermal stress on epoxy resins, showing a good fit with the experimental results. However, above 80 °C, Fick’s second law hygrothermal analogy quickly fails. The study by Fan et al. [29] explains this phenomenon; at actual temperatures below the polymer glass transition temperature *Tg*, the saturated moisture concentration *C_sat_* is only related to the relative humidity, whereas, when the actual working temperature reaches or even exceeds the glass transition temperature, the saturated moisture concentration *C_sat_* becomes rapidly larger as the temperature rises. This is because the diffusion of moisture in polymers above the glass transition temperature *Tg* is heavily dependent on temperature changes. In this regard, Wong et al. [30] found that the activation energy for solubility of plasticized materials ranges from 44,000 to 46,000 J/mol, which suggests that the saturated wet concentration *C_sat_* is a function of temperature above 100 °C [19]. Bao et al. [31] gave a functional relationship between *C_sat_* and temperature:(3)$$C_{sat}=C_{sat,ref}\exp\left[-\frac{\Delta H_{abs}}{R}(\frac{1}{T}-\frac{1}{T_{ref}})\right]$$
where $C_{sat,ref}$ is the saturated moisture concentration at a reference temperature $T_{ref}$ and $\Delta H_{abs}$ is the net heat of moisture absorption of the material. Additionally, since the temperature varies with time, the saturated moisture concentration *C_sat_* is also a function of time [32]. Various models have been proposed to solve this problem. For example, Jang [33] developed the “advanced normalization method”, but this requires that the solubility activation energies of the materials at the adjacent interfaces are equal. Khalilullah [34] used the two-stage diffusion model proposed by Placette [35] to simulate the diffusion of water vapor inside polymers:(4)$$C(x,t)=C_{1}(x,t)+C_{2}(x,t)$$

Since the two stages employ Fick diffusion superimposed with no-Fick diffusion to form the total humidity concentration, it is characterized by an initial wet diffusion that rapidly reaches a quasi-equilibrium state followed by a slow hygroscopic absorption to approach the final equilibrium state, and its accuracy is questionable. Wong [36] introduced an “endogenous technique” to compensate for the omission of the temperature term in the finite element calculation by introducing an endogenous term *r* to compensate for the omission of the temperature term in the finite element calculation.
(5)$$\frac{\partial w}{\partial t}=D\left(\frac{\partial^{2}w}{\partial x^{2}}+\frac{\partial^{2}w}{\partial y^{2}}+\frac{\partial^{2}w}{\partial z^{2}}\right)-\frac{w}{C_{sat}}\frac{\partial C_{sat}}{\partial t}$$
where r is defined as:(6)$$r=-w\frac{\partial C_{sat}}{\partial t}$$

However, this is highly dependent on the number of iterations in each time step. For this reason, Park [37] reworked the “endogenous technique method” to eliminate the iteration requirement and make it simpler. J. Wang [38] subsequently verified the reliability issue of the endogenous technique method and found that it has a good fit with the experimental results. In addition, Diyaroglu et al. [39] proposed a weekly dynamic humidity model based on near-field dynamics:(7)$$C_{sat}\frac{\partial w_{\left(k\right)}}{\partial t}=\sum_{j=1}^{N}f_{\left(k)(j\right)}(w_{\left(j\right)},w_{\left(k\right)},x_{\left(j\right)},x_{\left(k\right)},t)V_{\left(j\right)}+\theta_{m(k)}$$
defining:(8)$$\theta_{m(k)}=-w_{\left(k\right)}\frac{\partial C_{sat}}{\partial t}$$
where $w_{\left(j\right)},w_{\left(k\right)}$ is the humidity value at $x_{\left(j\right)},x_{\left(k\right)}$ respectively, and the response function $f_{\left(k)(j\right)}$ represents the humidity exchange between $x_{\left(j\right)},x_{\left(k\right)}$ points. Since the peridynamic humidity method does not contain any spatial derivatives to obtain the interface continuity conditions, this model can accurately and conveniently calculate the internal humidity concentration of complex structures with different material encapsulants for changing time and temperature.

To summarize, when it is necessary to determine the degree of moisture absorption of a single electronic packaging material at room temperature and at low temperatures, Fick’s second law hygrothermal analogy method exhibits (1) wide applicability and (2) computational simplicity to fit the internal moisture diffusion of the packaging material well. However, it is difficult to accurately characterize the composite kinetic boundary condition diffusion problem using the hygrothermal analogy when the discontinuity of the encapsulated material and a high diffusion ambient temperature are present, showing its limitations. The newest wet diffusion model, the “peridynamic humidity model”, and the “endogenous technology method”, can well characterize the diffusion of moisture in complex electronic packages at different temperatures, but they also have certain different defects that need to be addressed according to the actual situation in the practical application (Table 1). The trade-offs (Table 1) for weekly dynamic humidity modeling are as follows:The model introduces time and space factors to better match the actual diffusion process;The model describes wet diffusion inside complex packages and in two-phase boundary layers at the microscopic molecular level;The model requires a large computational and experimental effort to determine the model parameters;It is difficult to accurately describe the wet diffusion process of the whole macrostructure in a small part of the selected microstructure model.

For the endogenous technique method:An endogenous term *r* is introduced to compensate for the omission of the temperature term in the advanced normalization method;Specific modeling is required for specific problems, with poor generality and macroscopic regularity;There is higher computational complexity.

From the above analysis, it can be concluded that the peridynamic and endogenous methods provide high accuracy and fitness with increased computational complexity and model specificity. Therefore, when describing the wet diffusion process, Fick’s second law hygrothermal analogy can be used first to predict the diffusion trend, and then the peridynamic or endogenous methods can be used to further optimize the parameters. This strategy of combining different methods can make up for the shortcomings of various models and achieve an efficient and accurate description of the wet diffusion process. In addition, the continuous simplification of wet diffusion models is also an ongoing research direction to reduce their model complexity, which deserves attention.

**Table 1 micromachines-15-00376-t001:** Comparison of various wet diffusion models.

Wet Diffusion Models	Theoretical Foundations	Improvement Aspects/Problems
Thermal diffusion analogy [28]	*Fick’s* Second law of diffusion	Discontinuities in interfacial parameters
Normalization method [27]	Introduction of normalized concentration *W*	Solve the discontinuity of interfacial parameters
Advanced normalization method [33]	Introduced temperature *T* as a parameter	Solve the problem of C_sat_ variation with time
Endogenous technique method [37]	Introduce endogenous terms *r*	Solve the neglect of the temperature term in FEA
Peridynamic humidity method [39]	Introduces near-field dynamics High accuracy	Wide range of applicability, but high requirements for experimental equipment

#### 2.1.2. Physical Damage by Moisture

In the previous explorations of the mechanism of delamination failure of plasticized devices, the main focus is on the mismatch of the CTE (coefficient of thermal expansion) of different materials to explain the delamination phenomenon by the exacerbation of thermal stresses [10,40,41]. However, in addition to the thermal strain caused by thermal expansion of the device, there are also wet strains caused by the mismatch of the CHS (coefficient of hygroscopic swelling) of various materials inside the device due to the difference in hygroscopic capacity, as well as vapor pressure strain caused by the conversion of water molecules in the sealing material into water vapor due to heat.

Nguyen et al. [42] tested the piezoresistive sensor test chip at 65% RH, 65 °C; 85% RH, 85 °C; and 95% RH, 95 °C; the study found that the maximum mass increment was up to 139 mg and the maximum stress was 37 MPa. Dong et al. [43] utilized an electro-thermal–wet coupling model to realize stress coupling under thermal cycling and wet expansion, and the results showed that the coupled stress of the silicon chip increased by 2.6 MPa to 26.361 MPa after the introduction of wet expansion compared to thermal stress alone (Figure 3a). Wang [27] chose the virtual crack closure technique (VCCT) to introduce the effect of CTE–CHS–vapor pressure superposition on the delamination of two-phase interfaces for simulation; it was found that the strain energy release rates produced by hygroscopic loading *G_h_* and vapor pressure loading *G_p_* were similar in magnitude to those produced by thermal loading *G_t_* (Figure 3c,d). This result suggests that, in addition to the effect of temperature, moisture and vapor pressure also play an important role in delamination in stacked mold encapsulation. He [28] applied an experimental method to determine the hygroscopic swelling behavior of EMCs using a dynamic mechanical analyzer with a moisture generator (MDA-RH system) at 30 °C, 60 °C, and 80 °C temperature conditions, as well as 60% RH humidity. The results are shown in Figure 3b. In particular, at 60 °C and 60% RH, it is observed that the strain due to hygroscopic expansion of EMCs is equal to the strain due to thermal expansion at 0~55 °C or 150~175 °C.

The above studies show that the total strain induced by thermal expansion, moisture expansion, and vapor pressure is the main cause of delamination failure at the two-phase interface of the device under humid–thermal conditions in the true sense of the term, and the induced strains are all in the same order of magnitude. In heterogeneous integrated packages, the initial expansion of the crack at the edge position is mainly caused by shear stress, while the center position is mainly caused by normal stress. The larger the initial crack, the greater the stress contribution from vapor pressure, leading to a slow increase in the total strain energy release rate. With the increase in the initial crack length, the crack phase angle gradually decreases, indicating that the stress leading to crack extension is gradually converted from shear stress to normal force. In addition, the strain energy release rate gradually increases, the effect of hygrothermal stress gradually decreases, and the effect of steam pressure gradually becomes the main factor. With the gradual expansion of the cracks, the vapor pressure accumulates in the cracked region, which makes the effect of vapor pressure decisive for the final delamination failure of the device. This is the final macroscopic manifestation of the device’s “popcorn” effect.

#### 2.1.3. Chemical Damage by Moisture

The non-airtightness of the plastic seal causes the intrusion of a large number of non-polar molecules and polar molecules along the micropores, including O_2_, which can undergo oxidation reactions, and H_2_O, which can undergo corrosion reactions [44,45]. One of the well-established mechanisms in the research field is the oxidation of the copper substrate in the plasticized copper substrate or the RDL (ReDistribution Layer) in the presence of O_2_ and water molecules carrying Na^+^, Cl^−^, and other impurity ions through the chip passivation layer, and the metal-layer corrosion reaction [13,14,15,16,17,18,46,47,48], as shown in Figure 4. However, the effect of thermal oxidation of epoxy resin in the presence of O_2_ and H_2_O on its adhesion is less mentioned.

In 1972, Sharpe proposed the interphase concept to describe the transition region at the interface between two phases of different materials [50]. At the EMC/Cu interface, when the oxide thickness of the Cu substrate is in the range of 20–30 nm, the increase in surface roughness leads to the enhancement of the binding at the two-phase interface. However, when the oxide thickness exceeded 50 nm, the lattice mismatch at the two-phase interface of CuO/Cu_2_O was the main reason for the decrease in the bonding (Figure 5). In addition, the preferential cross-linking reaction between the Cu substrate and the curing agent results in a lower cross-linking density of the EMC boundary layer, which is also an important reason for the decrease in the binding strength. The poorer wettability also leads to the decreased binding of Cu with organic polymers [51]. Meanwhile, at the interface of the EMC-chip passivation layer, water molecules continue to diffuse inward with thermal movement during the continuous moisture absorption process. When reaching the EMC-chip two-phase interface, the water molecules near the EMC side form hydrogen bonds with the polymer, which easily forms a weak boundary layer [52].

In terms of exploring the mechanism of polymer thermal oxidation, Zhao et al. [53], through high-temperature storage (HST), temperature cycling (TCT), and pressure cooking (PCT) experiments (Table 2), used Attenuated Total Reflection Fourier Transform Infrared Spectroscopy (ATR-FTIR) to analyze the EMC/Cu fracture surfaces, and found that under the action of copper with water molecules and oxidants, the polymer aliphatic C-H bond chain dissociated to form hydrogen peroxide species. The authors concluded that it is copper as a catalyst that catalyzes the high-temperature chain dissociation of hydrogen peroxide, resulting in the breakage of the polymer chain and the decrease in the bonding force of the final epoxy plastisol (Figure 6). This corroborates some of the literature that suggests that copper ions/copper oxide can act as a catalyst to accelerate the catalyzed thermal oxidative decomposition of polymers at high temperatures [54,55]. In addition, Yan et al. [56] analyzed the aging behavior of epoxy resins under the experimental conditions of humid–thermal cycling. It was shown that the property changes of epoxy resin during thermal cycling have obvious stages. In the initial stage of thermal cycling, the epoxy resin undergoes thermo-oxidative cross-linking, and its cross-link density and Shore hardness gradually increase. The structure of the resin at this stage becomes more compact. However, with the increase in the number of thermal cycles, the epoxy resin begins to show the phenomenon of thermal oxidative decomposition. At this stage, the glass transition temperature (*Tg*), thermal stability, and dielectric properties of the epoxy resin begin to decrease. This change not only affects the intrinsic properties of the material, but also leads to externally observable changes, such as the appearance of exterior failure phenomena such as surface cracks, and a decrease in Shore hardness (Figure 7).

Based on the above studies, the chemical destruction of the Cu/EMC interface is layered with four main failure mechanisms:Cu-catalyzed thermo-oxidative degradation: Cu acts as a catalyst involved in the thermo-oxidative decomposition of polymers. The aliphatic C-H bonds, H_2_O, and O_2_ in the microcavities in the polymer react under the high temperature catalyzed by Cu^+^/Cu to generate hydrogen peroxide, which appears as the destruction of the polymer cured cross-link chain.Boundary layer weakening phenomenon: (1) The curing agent’s polar functional groups compared to the resin in the hydroxyl group are often preferentially adsorbed on the surface of the Cu metal, resulting in a decrease in the cross-linking density at the interface of the plastic sealing material. (2) At the same time, the water molecules at the interface pores and the chip passivation layer compete for hydrogen bonds with the polymer molecules in the molding material. This competitive chemical binding eventually leads to the emergence of a weak boundary layer phenomenon.Cu/CuO/Cu_2_O interfacial mismatch: Interfacial mismatch is mainly CuO/Cu_2_O intersecting interfacial lattice mismatch, resulting in a decrease in mechanical interlocking force.Surface tension, wettability, and hygroscopicity: The infiltration of the material is manifested as Cu < Cu_2_O < CuO; poor wettability will lead to a decline in the bonding of Cu and organic polymers.

These four chemical delamination failure mechanisms are superimposed on each other to cause damage to the outer appearance of the delamination interface, which thus shows a mixed failure state. For example, a portion of the epoxy encapsulant and its cleavage remains on the rough copper substrate; chip passivation layer peeling and epoxy encapsulant cracking coexist.

### 2.2. Damage by Thermal Stress

Compared with moisture damage, the thermal stress damage mechanism has been more studied. At present, the thermal stress damage mechanism generally recognized by academics mainly originates from the mismatch of the CTE of different materials inside the encapsulated device. This mismatch in the high temperature and temperature cycle of the operating environment, resulting in a variety of materials due to the deformation of the distance and direction of the different internal stresses, cannot be eliminated and balanced. When these accumulated stresses exceed the elastic yield strength of the material, device failure may be triggered. In the process of deepening the study of the thermal stress damage mechanism, researchers have formed a basic consensus that thermal stress damage mainly involves three key factors: CTE mismatch, thermal cycling stress, and external mechanical stress.

In terms of CTE mismatch, Conversion [57] utilized ANSYS simulation software to perform finite element simulation analysis of the EMC/Si chip interface adhesion force and the magnitude of maximum shear stress versus principal stress. The results show that the CTE mismatch of the EMC/Si chip induces a continuous increase in shear stress and principal stress, which eventually exceeds the EMC/Si chip interface adhesion force, and interfacial delamination is unavoidable. Chen et al. [58] also showed, in their research work on package warpage, that the wafer warpage can be significantly optimized by selecting EMCs with smaller CTEs and by reducing the thickness of EMC. Salahouelhadj [59], in a simulation of warpage inside FCBGA packages due to CTE mismatch of different material layers, found that the maximum warpage value of 67 μm occurs near 70 °C and 87 μm at 250 °C during heating. Subsequent warpage analysis also showed that FCBGA packages exhibit different concave curvatures in the chip region at the center of the package and in the region outside of the chip, resulting in a severe delamination phenomenon. Cherman [60] conducted CPI (Chip–Package Interaction) tests on two flip-chip joining processes (Table 3) using specialized test chips. It was shown that the mismatch between the cooling temperature difference and CTE (coefficient of thermal expansion) can lead to severe deformation under more severe MR temperature conditions. In addition, the authors’ study shows that the use of laminate substrates with a relatively low CTE (≤5 ppm/°C) also significantly reduces overall package internal stress. 

In terms of thermal cycling stress, Xue [61] conducted TC tests from −65 °C to 150 °C for flip-chip QFN 5 × 5 packages. After 240 temperature cycles, electrical failures gradually appeared, and the failure rate increased with the accumulation in the number of temperature cycling tests. Lau et al. [62] heterogeneously integrated a large chip and two small chips, and found that the maximum cumulative creep strains were all found at the corner solder joints of the chips through thermal cycling tests, i.e., any solder joint failures first appeared from these corner solder joints. Shie [63] investigated the damage mechanism of the *Cu- Cu* bumps under the thermal cycling test (TCT). After 1000 thermal cycles, the damage mechanism of the bonded *Cu- Cu* bumps under the thermal cycling test was investigated. After 1000 cycles, many cracks were found in the center of the bonding interface, which indicated that mechanical fatigue after thermal cycling was the main reason for the initial cracks. Under the thermal cycling condition, Chen et al. [64] investigated the delamination crack failure mechanism of UBM (Under Bump Metallurgy) and low-k inner layer dielectric (ILD) in FCBGA packages under temperature cycling conditions. The results show that the thermal stress caused by the CTE mismatch of different interface materials is the main reason (Figure 8). Additionally, the generated cyclic thermal stress may cause the metal layer with weak adhesion on the chip surface to slip during thermal cycling, resulting in a metal-layer short or open circuit failure; it may also lead to the rupture of the passivation layer or polycrystalline silicon layer, as shown in Figure 9, resulting in the short circuit between the multilayered metallization layers [65]. In addition, RDL delamination, an important component of advanced packaging under heterogeneous integration, also frequently occurs after temperature cycling tests, where fatigue stress at the material interface leads to cracking due to repeated thermal expansion and contraction of different material layers in the RDL. Similarly, delamination of the rewiring layer was also observed in impact testing [66]. Lau et al. [67] experimentally evaluated the characteristic lifetime of the rewiring layer and found that, at 0.0023 s after experiencing an impact, the maximum stresses of the package appeared in the four corners, with the bottommost rewiring layer undergoing the greatest stress and propensity to cracking. Similar stress distributions were also found in the study by Li et al. [20] (Figure 10).

In terms of external mechanical stress, Shih et al. [69] found that when FCGBA assemblies are subjected to thermal cycling, the extra weight of the heat sink exerts additional mechanical stress on the solder joints, resulting in the solder balls at the corners of the uncovered FCBGA assemblies being more susceptible to more severe cracking. To eliminate the negative effects of external vibration and shock on package reliability, researchers have proposed several ways to improve the electrical interconnect structure. The first, and most widely used, interconnect technology is the C4 bump. It uses a metal ball (usually a lead-tin alloy) to connect the chip PAD to the PCB, which absorbs thermal and mechanical stresses and improves shock and vibration resistance [70]. The second is the Cu pillar (Copper pillar); this is an emerging lead-free interconnect technology. In Cu pillar technology using copper pillars instead of a lead-tin ball, the mechanical strength is better but the shock resistance is also more likely to be enhanced [71]. The third type is the Ni/Sn pillar; this uses a stacked structure of nickel pillars and tin balls, which provides good stress relief and allows for finer spacing. This is also a lead-free packaging technology. The fourth is the Cu/Ni/Sn pillar, in which the Cu pillar is plated with a Ni layer, and then covered with tin balls. This not only retains the advantages of the high strength of Cu, but also has the function of Ni/Sn stress relief. Finally, the interconnecting structure filled with primer can significantly enhance the support and cushioning effect, and improve impact resistance [72].

In a study exploring the effect of thermal stress on the two-phase interface, we observe a remarkable phenomenon that, either due to the mismatch of the CTE of the package material or the superposition of thermal cycling, as well as external mechanical stresses, the package or the chip, under the action of thermal stresses, firstly shows a stress concentration at the corners, and then the stresses are gradually spreading from the edges to the center, which shows a gradual pattern of gradual progression from the outside to the inside. This situation suggests an important empirical conclusion that the stress concentration phenomenon is more likely to occur in the edge region due to the influence of a weak boundary layer. This finding is instructive for design practice. Especially in the package or chip design process, we should pay more attention to the stress concentration at the edges to optimize the design and prevent possible thermal stress damage.

### 2.3. Wrap-Up

In the analysis of the delamination phenomenon of specific plasticized devices, a large number of cases are based on experiments and finite element analysis simulation to jointly construct the failure mechanism of delamination, and concluded that the delamination phenomenon is the appearance of failure caused by the joint action of thermal stress and moisture, and mutual excitation. This shows that the delamination failure of the plastic sealing device is not a single factor under the action of a single result, nor is it a simple superposition of the results of the formation of a variety of factors under the independent action of the factors; rather, it is the thermal–electrical–force multi-physical field coupling under the action of a variety of factors interacting with each other to obtain the linkage results. For example, in order to simulate the package delamination failure process in real operating environments, Shih et al. [73] developed a three-dimensional computational model for humid–thermal loading conditions to evaluate the moisture diffusion, thermal stresses, and combined stresses of multi-chip WQFN packages under typical processing conditions and pre-treatment conditions. They investigated the fracture behavior of the EMC/Cu interface by combining the validated computational model with the virtual crack closure technique. It was found that the moisture absorption effect exacerbated the interfacial crack extension induced by thermal cycling. Wang et al. [27] also used the VCCT to investigate the effects of moisture absorption, vapor pressure, and thermal loading on the interfacial delamination of the mold encapsulation. The authors found that the strain energy release rate due to moisture adsorption and vapor pressure was similar to that of thermal loading, suggesting that moisture is also an important factor leading to delamination, a conclusion supported by He’s experimental study. He [28] measured the hygroscopic expansion behavior of encapsulants under different temperature and humidity conditions. The results show that the strain due to hygroscopic expansion is comparable to thermal expansion at 0~55 °C or 150~175 °C at 60 °C and 60% RH. Dong et al. [43] investigated the joint effects of temperature cycling and moisture expansion on silicon chips using the coupled electric–thermal–humidity model. The results showed that the stress of the chip increased by 2.6 MPa when moisture expansion was taken into account. This indicates that moisture exacerbates the stress due to temperature cycling and is a non-negligible main driver of delamination failure. Based on the above conclusions, in the analysis and study of the delamination mechanism of plastic packaging devices, we should also turn to the appearance delamination based on multi-physics coupling and multi-factor interaction, so as to provide reliable and effective theoretical guidance for delamination failure in actual production and life.

## 3. The Most Recent Solutions for Delamination Failure

From the simplest molded packages to advanced packages based on heterogeneous integration, the most serious phase interfaces where delamination occurs are molded EMC/Cu substrate, EMC/Si chip interface, and PI/Cu wiring in the RDL. The reason for this is that H_2_O and O_2_ can enter the package through the polymer-based epoxy plastic sealer or polyimide (PI) to undergo corrosion and oxidation reactions, which together with the pre-existing thermal stresses cause hybrid delamination failure at the polymer interface. Currently, there are two ways to solve the problem of delamination failure: (1) improving the production process technology; and (2) changing the nature of a certain structure or material of the device package.

The first solution path is the most widely used. For example, adding a baking process to remove moisture before the reflow process after the device has been molded can be a good way to reduce delamination failures [74,75,76]. Addition of deionized water treatment prior to each step of the molding process is used to remove impurity elements introduced during the process. Ar+O_2_ plasma treatment is applied to the copper wiring of the RDL (Figure 11) to appropriately increase the roughness and hydrophilicity of the copper wiring [68] (hydrophilicity in this case refers to the increase in the wettability of the material in order to maximize the contact at the interface of the two phases, thus achieving an increase in the adhesive strength; Pai’s study also showed that the high hygroscopicity of the copper lead frames after plasma treatment and the high electric field gradient after energization hindered the interfacial adhesion and led to composite delamination [77]). A mixture of N_2_ and a small amount of H_2_ is added to some high-temperature processes to protect critical devices from oxidation. These changes to the production process can, to a certain extent, improve the likelihood of delamination failure of packaged devices. However, the failure paths of the devices are not completely isolated, so large-scale delamination failures will still occur in subsequent harsh dynamic service environments.

The second solution path is to change the structure of the package to realize the complete isolation of the device failure path, which results in a decisive improvement in the delamination failure. However, such structural changes often result in the degradation of other device properties such as thermo-mechanical properties, low dielectric properties, and CTE matching. Therefore, in the case of delamination failure, we should also take into account the impact of other package performances [78]. Nowadays, the more cutting-edge anti-delamination structural changes are mainly focused on five aspects: (1) adding a thin film layer on the surface of the RDL; (2) optimization of polymer fillers; (3) introduction of second-phase modifiers for EMC; (4) conjugated polymer self-healing technology; and (5) interfacial buffer layer technology. Among these, (1) is applied to isolate the moisture and small molecule oxygen immersion path in order to achieve the requirements of airtightness; (2–4) are used to improve the encapsulated device thermal conduction and stress–strain absorption; and (5) is used to improve the two-phase interface CTE mismatch by adding a buffer layer of new materials.

### 3.1. Addition of Barrier Layer on Top of RDL Stacks

The RDL (redistribution layer) is a widely used layer structure in IC packaging for electrical connection and signal transfer between the chip and the package substrate. The RDL is generally composed of polyimide PI, a diffusion-resistant Ti layer, and Cu wiring. Due to the CTE mismatch between PI/Cu and the non-gas-tightness of PI, which can allow H_2_O and O_2_ to pass through, the final delamination of the RDL is caused under dynamic service environments [79,80].

As shown in Figure 12a, the current mainstream RDL antioxidant technology is to use low-temperature atomic layer deposition (ALD) to deposit a layer of inorganic film on the copper wiring to block the entry of oxygen [81,82,83]. For example, Pinho [84] describes a process to integrate an oxygen diffusion barrier into the top of an RDL stack using atomic layer deposition (ALD), which exhibits a good barrier effect and can prevent Cu oxidative delamination. In addition, Chery [85] conducted an experimental comparison of several inorganic coatings based on low-temperature ALD technology—AI_2_O_3_, HfO_2_, and TiO_2_—in terms of their ability to block O_2_ and H_2_O from entering the interior of the RDL layer. As shown in Figure 12d,e, the 12 nm HfO_2_ inorganic coating performs the best, and the coating not only protects against the oxidation of O_2_, but is also completely unaffected by the presence of H_2_O in the corrosion stress test and is a better inorganic coating than AI_2_O_3_ in terms of performance indicators. However, AI_2_O_3_ is the preferred inorganic coating in terms of low-temperature manufacturability as a metric consideration. Specific comparisons are shown in Table 4. In addition, Wu et al. [86] implemented self-aligned fiber connections on a monolithic silicon photonic chip and achieved chip-level hermetic packaging by introducing a moisture barrier at 300 °C via the ALD technique. The authors found that, without the protection of a moisture barrier, the coupling between the fiber and the chip is affected by the humidity-induced increase in the refractive index of the oxide cladding. In contrast, with the introduction of a moisture barrier at 300 °C by ALD technology, the fiber optic coupler showed excellent performance in overstressing and testing. The cumbersome and expensive nature of ALD technology as an actively investigated solution for oxidative delamination failures has limited the large-scale popularization of ALD. The low cost of polymer-based thin-mold layers has attracted the attention of researchers. Moreover, this polymer-based thin-mold layer manufacturing process can be integrated into the RDL manufacturing process with greater simplicity than ALD technology. Chery [87] considered that Cu oxidation at high temperatures is the main factor contributing to RDL delamination. Firstly, eight polymers and one liquid mold material were used I the experiment as barrier layers at the top of RDL stacks, and it was found that the barrier layers could block the O_2_ immersion well, especially the liquid mold material. In a subsequent study, Kwak [88] applied these 200 um thick liquid-form layers to cover the photosensitive polymer layer to conduct a standard temperature and humidity (TH) stress test for 1000 h (test conditions 85 °C/85% RH. This showed that, for this RDL with a liquid-barrier layer, compared to the control group, no significant oxidation was observed for the 300 nm thick oxide layer. This shows that this liquid-barrier layer is a simple anti-oxidation layer technology that is easy to massively popularize.

The technology of stacking thin film layers on the top of RDL stacks is mainly considered to be anti-oxidation at present, and there is still a certain research blind spot regarding moisture protection. In material science research, the development of epoxy resin-based hydrophobic materials has shown excellent hydrophobicity and mechanical properties that deserve our attention [89,90,91], The application of hydrophobic materials or thin films of hydrophobic materials on electronic packaging can be tried in future research. In addition, polymer processing, in terms of temperature, requires the use of ALD technology below 120 °C to achieve low-temperature deposition. However, in low-temperature deposition of more complex surface reaction mechanisms, a slower growth rate is brought about by the inorganic coating densification, and barrier degradation is also a constraint to the development of inorganic coatings in the research. Compared with inorganic coatings, due to their low cost and simplicity, thin-mold layers of polymers are worthy of further research. However, the 200 μm thickness of their liquid mold layers creates a great challenge to the dimensional requirements of RDL rewiring layers.

### 3.2. Optimization of Polymer Inorganic Fillers

From plastic encapsulation to advanced packaging technology under heterogeneous integration, polymer has become an important material in the field of packaging due to its good thermosetting, plasticity, corrosion resistance, and low cost. However, the thermo-mechanical properties, electrical conductivity, and thermal conductivity of a single polymer are not well suited to the harsh working environment of electronic packaging. The use of polymer to add modifiers, such as inorganic fillers, can be very good approach to improving the overall electrical and thermal conductivity, enhancing the strength, stiffness, and wear resistance of the polymer, and, to a certain extent, reducing the coefficient of thermal expansion of the polymer, so that its CTE mismatch with the surrounding materials can be improved. However, determining how to add inorganic fillers and add different properties of fillers to show the difference in polymer properties has become one of the main directions of research.

#### 3.2.1. Diversification of Filler Particle Size

Wan et al. [92] proposed a correction to Einstein’s viscosity equation for particle-containing liquids based on Kreigher and Dougherty for the bottom filler, as shown in Figure 13a,b. The authors concluded that the filler particle size and distribution are the two most important factors in determining the concentration of the system. In terms of filler particle size, Wen [93] found that submicron fillers have a smaller specific surface area and surface energy compared to nanoscale fillers. Additionally, showing weak interaction with the matrix is also less prone to the formation of the agglomeration phenomenon, which is conducive to an increase in mobility. The mixing of different sizes of binary SiO_2_ and AI_2_O_3_, as shown in Figure 13d–f, increases the theoretical maximum stacking amount as the thermal conductivity and mobility increase. In a practical application, Guo et al. [94] prepared a SiO_2_ filler with bimodal size, as shown in Figure 13c. The two sizes of SiO_2_ interpenetrated with each other, reflecting excellent fluidity with higher *Tg*. Zhang [95] mixed 50% micron-sized silver-plated copper powder with 50% submicron-sized silver-plated copper powder as a conductive filler, and combined it with epoxy resin to form a new type of ECA material, which formed an excellent conductive pathway due to the interpenetration of the silver-plated copper powders of different particle sizes. Furthermore, Feng et al. [96] designed ternary S-AI_2_O_3_ particles according to the Dinger–Funk model that exhibited better thermal conductivity and mobility than binary blends. Under the requirement of balancing thermal conductivity, CTE matching, and fluidity, Lee et al. [97] designed a spherical AI_2_O_3_ and flake BN hybrid filler to achieve lower viscosity, higher thermal conductivity, and a moderate CTE value. A specific comparison is shown in Figure 14. 

In addition, the thermal conductivity of thermal interface materials with anisotropy is pursued by studying oriented fillers. The main principle is to allow the lamellar fillers to be ordered in a three-dimensional framework to achieve anisotropic thermal conductivity [92]. This ordered arrangement of a flat orientation greatly increases the sliding friction between the fillers, which in turn hinders the directional sliding of the polymerized molecular chains and also improves the overall mechanical strength [98]. The main fabrication methods include hot pressing, vacuum-assisted filtration, and freeze casting.

#### 3.2.2. Nanographene Fillers

In the field of inorganic fillers, solving the problem of filler dispersion in order to eliminate stress concentration and improve performance uniformity has been the focus of researchers. Among the relevant factors, the filler particle size is one of the key indicators used to determine the dispersion uniformity. In recent years, it has been found that the addition of nanoscale fillers can effectively solve this problem; only a small addition can significantly improve the performance of the overall system and the change in the density of the polymer matrix is not obvious. Since 2004, when graphene material was first mechanically exfoliated, its demonstrated high mechanical strength (1600 GPa), high specific surface area (2600 m^2^/g), and high thermal conductivity (5000 W/mK) have led the way for new developments in the field of materials science. Graphene is mainly categorized as graphene oxide (GO), reduced graphene oxide (rGO), and graphene nanoplatelet (GNP)/thin-layer graphene (FLG). It is worth noting that the combined use of nanoscale fillers and graphene shows great potential for application in the field of fillers, which undoubtedly provides new research directions and practical paths for solving the dispersion problem of inorganic fillers, eliminating the stress concentration, and improving the uniformity of properties. Huang et al. [99] determined the thermal conductivity of the composite to be 3.45 W/mK for the case of BN-rGO filler with a loading rate of 26.04 Vol%, which was 16 times higher than that of the pure epoxy resin. The ABS/GNP nanocomposites prepared by Dul et al. [100] exhibited excellent mechanical and thermal properties due to the incorporation of 4 wt% nano-GNP with little change in *Tg*. Tambrallimath [101] designed PC-ABS/GNP nanocomposites with 0.8 wt% GNP nanofillers to improve the tensile strength of the PC-ABS matrix by 57% and impact resistance by 87%. The GO-FGO/AA nanocomposites designed and fabricated by Dong et al. [102] have a maximum toughness at 0.04 wt% of GO and FGO content. These show that very small amounts of graphene nanofillers can change the properties of composites to a large extent. The result shows extremely good application prospects in the field of electronic packaging. However, graphene nanofillers for electronic packaging applications have certain drawbacks:The problem of uniform dispersion of graphene filler in polymer matrix still exists.The new graphene encapsulation material has strong thermal conductivity, thermo-mechanical properties, and thermal stability, and, to some extent, it enhances the anti-delamination performance of encapsulated devices. However, the interfacial adhesion between the composite material and other interfacial materials, and whether graphene oxidative degradation occurs with high-temperature thermal cycling, causing adhesion to decrease, all need to be further studied.The production of graphene requires a high energy input, but the output is small. Further optimization is needed in terms of production cost and scaling up.

#### 3.2.3. Filler Surface Modification

Filler surface modification is the process of changing the surface properties and interactions of a filler such as SiO_2_ by physically or chemically removing or shielding the -OH groups on its surface (Figure 15). This process can improve the filler–polymer interface interaction, increase the *Tg*, of the material and reduce the overall polymer thermal stress [103]. It is mainly categorized into physical adsorption modification and chemical modification (a specific comparison is shown in Table 5). Chemical surface modification in the presence of water can reduce the viscosity and also reduce the CTE of the plastic sealing material, which is the best preparation method for filler surface modification [104,105,106]. In practical applications, surface wettability and uniform dispersion of BN-type fillers can be improved by surface modification. Uncured epoxy resins filled with modified B-BN had the lowest viscosity at all shear rates compared to untreated BN fillers and conventional NaOH chemically modified BN fillers [107]. In addition, Peng et al. [88] used γ-aminopropyltriethoxysilane for surface modification of AlN nanofillers to prepare nano-sized AlN homogeneously dispersed epoxy composites with good dispersion and wettability compared to ordinary non-surface-modified polymer materials, thus showing its good prospects for application.

Diversification of filler particle size, application of nanoscale graphene fillers, and surface modification of fillers can well improve the compatibility and interaction between fillers and the polymer matrix in order to improve the overall polymer integrity and thermal conductivity for the purpose of anti-delamination. However, there are still some aspects of filler optimization that need further in-depth study: (1) development of easier and more effective chemical treatment-based surface modification technology; (2) the surface modification of the two-phase interface effect of the deep mechanism, in order to reveal the filler surface modification of the actual filler performance of the mechanism; (3) the mechanism of the interactions of the filler and the polymer matrix interface, which further allows variable regulation of interfacial structure, interfacial energy, and mutual diffusion; and (4) the homogeneous dispersion of nanoscale graphene fillers in the polymer matrix; the surface modification of nanoscale graphene fillers can be an important optimization direction to improve its homogeneous dispersion [108].

### 3.3. EMC Introduces Second-Phase Modifier-Block Copolymers

Pure epoxy resins are widely used for their excellent mechanical properties, chemical resistance, and adhesion. However, as encapsulation materials, inorganic fillers need to be introduced as modifiers to reduce the filler expansion coefficient and stiffness, and to hinder epoxy curing shrinkage in order to improve their heat resistance and toughness. However, the use of traditional inorganic fillers such as SiO_2_ and AI_2_O_3_ to obtain composites with excellent encapsulation properties is limited. Therefore, the introduction of second-phase modifiers to continue to improve the polymer matrix properties on the basis of inorganic filler modifiers has attracted the attention of researchers.

In early studies [109,110], carboxyl end-seal tougheners and amino end-seal tougheners were widely used to improve the mechanical properties and fracture toughness of epoxy composites in order to enhance the delamination resistance of the composites. However, Marouf [111] showed that these conventional tougheners reduce the *Tg*, viscosity, and flexural strength of composites, and are not suitable for electronic packaging material applications. Silicone rubber (SR) modifier is also one of the most effective polymer modifiers for delamination resistance, and a study [112] showed that polymer materials with 2 wt% SR addition exhibited excellent delamination resistance after 1000 thermal cycles. However, the same SR addition reduces the flexural strength and modulus of the polymer matrix, which requires further research for improvement.

In recent studies on electronic packaging materials, block copolymers have been proposed as second-phase modifiers to toughen the polymer matrix, and a triblock copolymer BCP: M52N in a PMMA-b-PBuA-b-PMMA structure was introduced. In it, DMA units were introduced into the PMMA chain (PMMA: poly(methyl methacrylate); PBuA: polybutyl acrylate; PDMA: poly(dimethylacrylamide)), and the flexural strength and toughness reached 152 MPa and 0.79 J at 83 wt% SiO_2_ and 1.5 wt% PDMA copolymerization BCP loading (Figure 16). Its optimal delamination resistance index was 74, demonstrating excellent delamination resistance (Table 6). Amino end-capped PCBs prepared by Li et al. [113] as a second-phase modifier increased the fracture toughness of EP/Amino-BCP/SiO_2_ at room temperature by 31.2% in the presence of 60 wt% SiO_2_; CTE1 increased by 6.1 ppm/K; and CTE2 was largely unaffected. Through comparative experiments, Yang et al. [114] found that amino block copolymers and silicone block copolymers showed a good toughening mechanism as toughening agents for bottom filling.

Block copolymers can be used as second-phase modifiers to improve the mechanical strength, thermal stability, and delamination resistance of polymers due to the interaction of their hydrophilic and lipophilic regions with polar/non-polar functional groups to form phase-separated microstructures in the polymer matrix, respectively. In addition, as second-phase modifiers, they can also be used to avoid the degradation of polymer properties caused by the unassembled free molecules produced by BCPs as modifiers alone. Although block copolymers as second-phase modifiers have good application prospects in the field of electronic packaging, there are many problems that need to be solved:The synthesis process of BCPs needs to be further optimized and simplified to achieve precise control of the microphase separation structure of BCPs.Whether the interfacial compatibility and anti-delamination performance of BCPs can be sustained under extreme conditions in electronic packaging remains to be investigated.The high cost of BCPs is a major obstacle to their large-scale application in electronic packaging, and there is a need to expand the synthesis of BCPs from laboratory processing to an industrial processing scale.There is less research on the co-toughening application of BCPs as a second-phase modifier and inorganic filler. Although it has been shown that the anti-delamination index of the polymer modified by 1.5 wt% BCP and 83 wt% SiO_2_ composite reached 74, the reduction in its fluidity and interfacial adhesion can cause the formation of other defects, such as plastic sealing, punching wire, and cavities.

### 3.4. Application of Conjugated Polymer Self-Healing Technology

Conjugated semiconducting polymers are a class of polymeric materials with a special structure containing a conjugated π-electron system in the molecular chain. The supramolecular interactions and dynamic bonding realized by this conjugated structure endow the polymers with good self-healing properties. When subjected to external stress, the internally stabilized and ordered structure provided by the supramolecular interactions of such self-healing polymers endows the polymers with stability in shape and properties under strain. When the stress of the attack is further increased, various forms of dynamic bonding, such as hydrogen bonding, metal coordination, and π-π stacking, release the material from the strain by breaking, and then the dynamic bonding is subsequently re-activated to achieve self-healing properties.

Ocheje et al. [115] designed a self-healing conjugated polymer with good mechanical properties by adding amide compounds. The added 10 mol% amide side chain exhibited a maximum 75% tensile elongation of the overall polymer due to the incorporation of hydrogen bonding. Moreover, this conjugated polymer can repair micrometer-scale damage under post-treatment of exposure to chlorobenzene and thermal annealing. Yu et al. [116] prepared a self-healing DPP-based conjugated copolymer, PDPP4T-DCM, utilizing a dynamic covalent bond breaking–bonding mechanism. The crack depth healed from 120 nm to only 4 nm under post-treatment such as UV irradiation (Figure 17a). The semiconductor hybrid film designed by Oh et al. [117] utilized the metal–ligand bonds formed between Fe^3+^ and N and O atoms to achieve a high tensile elongation, as well as a self-healing function with a dynamic cross-linking structure (Figure 17b). This self-healing function realized by metal–ligand bonding can be achieved at room temperature without post-treatment, but the time to achieve self-healing is more than 24 h. To achieve a short self-healing time at room temperature, Lenny et al. [118] designed self-healing conjugated polymers with PThP as the backbone and hydrogen bonding as the side chain. Results showed good mechanical and electrical recovery, and self-healing at room temperature in less than one minute (Figure 17c,d). When doped with other polymers, the copolymers showed good mechanical properties, but their elastic modulus decreased by a certain amount.

Currently, this technique of applying supramolecular interactions and dynamic bonding to realize polymer self-healing is mainly used in flexible electronics, such as FETs acting as semiconductor channel layers, gate electrolyte layers, and control electrodes. It is foreseeable that conjugated polymers, with their excellent self-healing properties, will also have good application prospects in advanced packaging materials for heterogeneous integration. These applications are mainly reflected in the following:Different shapes of electronic devices need flexible packaging materials to provide protection and support functions;At the two-phase interface, conjugated polymers can provide a good stress buffer layer with excellent repair performance;Conjugated polymers can automatically repair any damage or malfunction in electronic devices to extend the service life of electronic devices.

However, there are also some difficulties in self-repairing conjugated polymers used in packaging anti-delamination technology:Thermal cycling ability: Conjugated polymers may fail in their self-repairing performance under the extreme working environment of high temperature and temperature cycling, and their thermal stability needs to be further improved.Interfacial compatibility: The bonding performance and interfacial compatibility of conjugated polymers as complex chemical materials with other materials under heterogeneous integration need to be further studied.Rigidity requirements: The dimensional changes of self-healing materials at high temperatures and humidity may also affect the reliability of electronic packaging. The dimensional changes of self-healing materials at high temperatures and humidity may also affect the reliability of electronic packaging.Maturity of self-healing technology: At present, hydrogen bonding, covalent bonding, and many other forms of dynamic bonding need to be applied as a specific post-treatment to achieve self-healing, whereas metal coordination dynamic bonding does not require a specific post-treatment, but needs to be left for more than 24 h to achieve self-healing. Further optimization of the self-healing technology is needed to achieve a short self-healing time of conjugated polymers at room temperature.

### 3.5. Application of Two-Phase Interface Buffer Layer Technology

Multiple advanced materials introduced for advanced packaging under heterogeneous integration show delamination tendency at the two-phase interface, mainly represented by shear stress and subsequent normal phase force, due to their own CTE mismatch and low interfacial adhesion when the two-phase interface encounters a high-temperature attack. In order to improve the reliability of advanced packaging technology and the matching degree of the CTE of materials at the two-phase interface, researchers propose to add a buffer layer at the two-phase interface to solve the compatibility problem between two-phase materials.

To better enhance the delamination-resistant reliability of third-generation semiconductor silicon carbide modules, the wide-bandwidth semiconductor material AIN can be used as a growth buffer layer between SiC and GaN, i.e., InGaN [120]. This is a reliable buffer layer material that can be applied on a large scale due to having similar lattice constants and high thermal conductivity to those of GaN and InGaN [121]. However, the complex preparation process of AIN limits its large-scale industrial production. Based on this, McLeod [122] designed a 580 °C pulsed CVD aluminum nitride process to simplify the process compared to the previous MOCVD, MBE, etc., and to obtain single crystals with quite good epitaxial relationship with the substrate, which greatly improves the crystallinity and density of the reaction-sputtered AIN. In addition, Tanaka et al. [123] designed a fatigue-free chip surface packaging technology based on the delamination failure of SiC and Al leads due to the CTE mismatch, which can realize 334,000 power cycles with a maximum junction temperature of *T_jmax_* of 200 °C and a temperature swing of *ΔT_j_* of 135 °C (Figure 18). The main technical point of this fatigue-free chip packaging technology is that a Cu-invar-Cu (CIC) CTE-matched buffer layer with a thickness ratio of 1:3:1 is bonded on top of the SiC using sintered copper technology. The “invar” is a special iron-nickel alloy. This CIC buffer layer reduces the overall CTE to 5 ppm/K, which improves the degree of CTE matching between SiC and copper leads [124] In a follow-up study, Shinkai et al. [110] realized variable CTE regulation of CIC by continuously adjusting the thickness ratio of the CIC buffer layer. Through comparative experiments, it was found that when the overall thickness of the CIC buffer layer was 0.2 mm and the thickness ratio was 1:18:1, the overall CTE was reduced to 2.1 ppm/K, and 905,000 power cycles were realized. It is also found that the invar interlayer with good stiffness can well impede crack propagation and reduce transient thermal impedance to enhance the delamination resistance of SiC modules. The design of the buffer layer also has some applications in other packaging technologies. For example, Li et al. [125] developed a novel aluminum nitride composite coating, AACC, as a buffer layer to effectively reduce the thermal mismatch between the chip and the heat sink W-Cu. The authors also mentioned that, by adjusting the ratio of AIN to AI, new composite coatings with variable CTE can also be formed. In addition, Liu et al. [126] designed a double-sided power module BM with copper wire spacers (CWSs) that extends the BM lifetime by 40.0% and 42.9%, respectively, over the double-sided BM with a conventional solid copper buffer layer. This exhibited excellent thermo-mechanical reliability in dynamic service environments.

The introduction of composite cushion layer designs has shown significant benefits in terms of absorbing thermal stresses and mitigating structural changes due to thermal deformation, in the face of mismatches in the coefficients of thermal expansion between the two interfaces of the different material phases and the need to improve delamination reliability. However, current uses, such as copper–iron–nickel alloy composite (CIC) buffer layers and aluminum nitride–aluminum (AlN-Al) composite buffer layers face challenges in moving from the laboratory environment to large-scale industrial manufacturing. These challenges mainly stem from the complexity of the preparation process and the need for fine control of the process steps. In order to overcome these challenges, the development and optimization of newly introduced encapsulation buffer layers is of particular importance. In this regard, we need to continuously explore new materials and preparation processes in order to simplify the production process and make it more adaptable to the needs of large-scale industrialized production. In addition, for the newly introduced buffer layer, we also need to comprehensively consider its interaction with other encapsulation materials and devices, and conduct global optimization design on this basis.

## 4. Summary

Delamination failure of polymer-based electronic packaging devices in dynamic service environments seriously affects the reliability of electronic devices. In this paper, the delamination failure mechanism at the two-phase interface of polymer-based packages is explored through a literature collation and analysis, and it is proposed that thermal stress damage and hygroscopic damage are the main intrinsic factors leading to delamination. It is found that the strain induced by hygroscopic expansion is comparable in order of magnitude to that induced by thermal expansion, which constitutes one of the important causes of delamination failure. In addition, it is further found that, in the presence of oxygen and moisture, Cu/Cu^+^ penetrates into the polymer matrix through the phase interface and catalyzes the thermo-oxidative chain decomposition of the epoxy resin, which is also an important factor contributing to delamination failure.

To solve this delamination problem, several recent solutions to isolate the delamination failure path by changing the original structure and materials of the package are summarized in this paper:Adding a thin film layer on top of the RDL stack. One approach is to apply ALD technology to deposit an inorganic barrier layer on the top of the RDL stack to isolate the intrusion of O_2_ and H_2_O, in order to hinder the oxidation reaction of Cu at high temperature. The polymer processing process of the RDL layer requires ALD technology to realize low-temperature deposition at 120 °C. However, the decrease in inorganic coating densities and barrier properties brought about by low-temperature deposition is a major research challenge that restricts the application of inorganic coatings to the RDL layer. In addition, the cumbersome and expensive steps of ALD technology also limit its industrial- and large-scale application. The second approach is to add a polymer-based thin film layer; a liquid mold material developed as a barrier layer on the top of the RDL stack shows good anti-oxidation and corrosion effects, but its 200 um thickness creates a great challenge to the overall dimensional requirements of the RDL layer.Filler optimization. Different shapes of submicron and nano-sized SiO_2_, AI_2_O_3_, and BN binary and ternary fillers have been used to improve the poor fluidity and easy agglomeration of the polymer. Among these, the addition of nano-sized graphene fillers can greatly enhance the mechanical and thermal properties of the overall polymer. However, the dispersibility of graphene fillers and the scale of production are problems that need to be further optimized. In addition, the use of filler surface modification can enhance the compatibility and wettability of the filler with the polymer surface, improve the thermo-mechanical properties of the material, and reduce the moisture absorption effect.EMC introduces a second-phase modifier. The 70% to 90% loading rate of inorganic fillers is limiting for composites with excellent encapsulation properties of the polymer matrix. Therefore, the introduction of second-phase modifiers on the basis of inorganic filler modifiers continues to improve the performance of the polymer matrix. Among these, block copolymers, in joint action with inorganic fillers, show excellent mechanical properties; due to the formation of block copolymers in the polymer matrix phase separation microstructure, the composite material of the anti-delamination index can reach 74. However, the addition of BCPs to the composite material reduces mobility and adhesion. Furthermore, the complexity of the BCP synthesis process is higher. These factors impede the large-scale application of BCPs in the field of electronic packaging.Self-healing technology. The supramolecular interactions and dynamic bonding realized by the conjugated structure of conjugated polymers endow the polymers with good self-healing properties. In terms of anti-delamination of electronic packaging, they can be used as stress buffer layers and flexible packaging materials to provide excellent protection and repair performance of electronic devices. However, the self-healing properties of such conjugated polymers require specific and complex post-treatments or a long standing time to achieve; therefore, further optimization is required to achieve a short self-healing time of conjugated polymers at room temperature. In addition, the conjugated polymers’ extreme working environment, self-healing properties, and adhesive properties also require further research and optimization.Buffer layer technology. The composite interfacial buffer layer is a reliable interfacial layer applied to alleviate the thermal mismatch at the two-phase interface, impede crack propagation, reduce transient thermal resistance, and improve interfacial delamination resistance. The AIN lattice buffer layer, CIC composite buffer layer, and the new nitrogen–aluminum composite buffer coating (AACC) all show excellent thermo-mechanical properties, alleviating the structural changes caused by the compatibility of the two-phase interface and thermal deformation. However, in terms of practical applications, the development of new materials and processes for cushion layers requires further simplification of the production process in order to adapt them to the requirements of large-scale industrialized production.

At present, these solutions represent great progress, but there is still a certain gap between these new anti-delamination solutions and technologies and their practical application. The main gap lies in the lack of mature technical routes for these new technologies and solutions to transition from laboratory preparations to large-scale industrial applications. In addition, the development of advanced encapsulation materials and two-phase interface micro-exploration technology, the optimization of the internal structure of the RDL, the development of special functional polymorphic polymers, and the use of a variety of strategies to coordinate the thermal imbalance and hygroscopic effects are expected to fundamentally solve the problem of delamination in polymer-based encapsulants, so that the reliability and service life of this type of encapsulant device can reach a completely new level.

## Figures and Tables

**Figure 1 micromachines-15-00376-f001:**
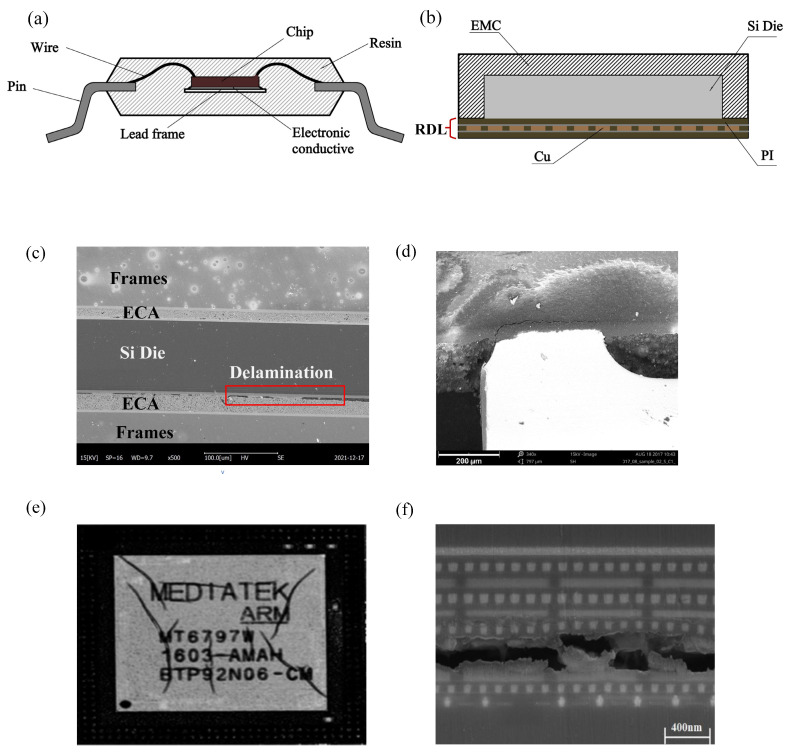
(**a**) SMD device schematic. (**b**) FO package device schematic. (**c**) A DFN-type package electronic device Die/ECA-layered SEM diagram. (**d**) Small cracks on the package body, next to a lead frame, are present on new components. Reproduced with permission [20]. Copyright 2021, Elsevier. (**e**) Cracks inside the chip after 500 cycles of thermal shock. (**f**) Cross-section of BEoL stack showing a horizontal crack at M9. Reprinted from Ref. [21]. with open access from MDPI.

**Figure 2 micromachines-15-00376-f002:**
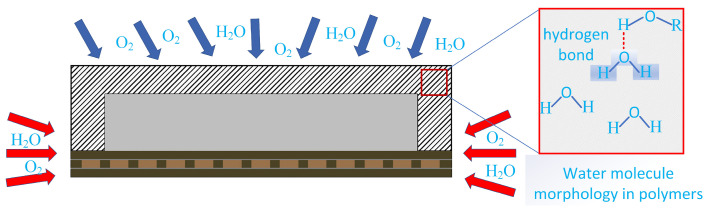
Polar/non-polar molecule intrusion paths into the package and the morphology of water molecules in the polymer.

**Figure 3 micromachines-15-00376-f003:**
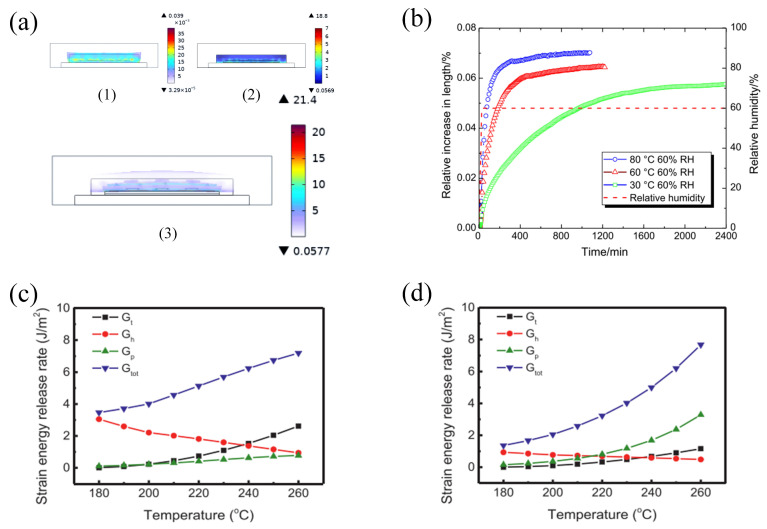
(**a**) Analysis of stress results for the silica gel layer (MPa). (1) Thermal stress. (2) Moisture stress. (3) Stress after thermal−moisture coupling. Reprinted from Ref. [43], with open access from MDPI. (**b**) Increase in sample length (hygroscopic swelling) versus time at 30, 60, and 80 °C and 60% RH. Reproduced with permission [28]. Copyright 2023, Springer. (**c**) Variation in G_t_, G_h_, G_p_, and G_tot_ with temperature at bottom die/die attach interface. (**d**) Variation in G_t_, G_h_, G_p_, and G_tot_ with temperature at bottom die/molding compound interface. Reproduced with permission [27]. Copyright 2020, Elsevier.

**Figure 4 micromachines-15-00376-f004:**
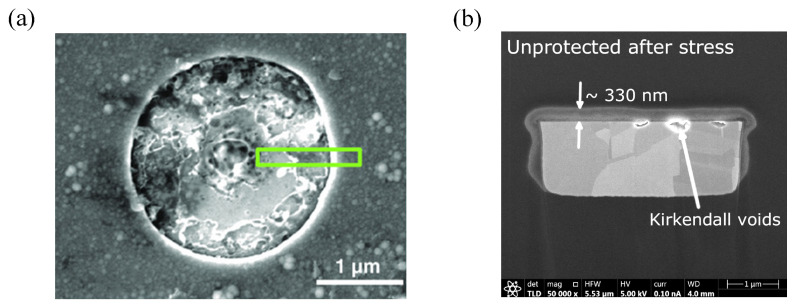
(**a**) Corroded region SEM images at the rear surface of a bifacial monocrystalline solar cell. Reproduced with permission [47]. Copyright 2019, John Wiley & Sons. (**b**) FIB cross-sections of a copper metal line encapsulated in polymer. After 1000 h spent at 150 °C, a 330 nm thick oxide layer is present at the top of the metal line and Kirkendall voids appeared at the copper–oxide interface. Reproduced with permission [49]. Copyright 2022, Elsevier.

**Figure 5 micromachines-15-00376-f005:**
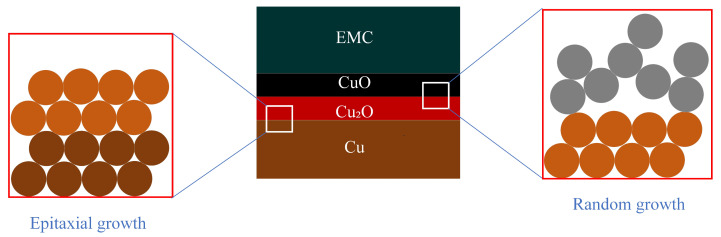
Epitaxial alignment growth of Cu_2_O and random alignment growth of CuO.

**Figure 6 micromachines-15-00376-f006:**
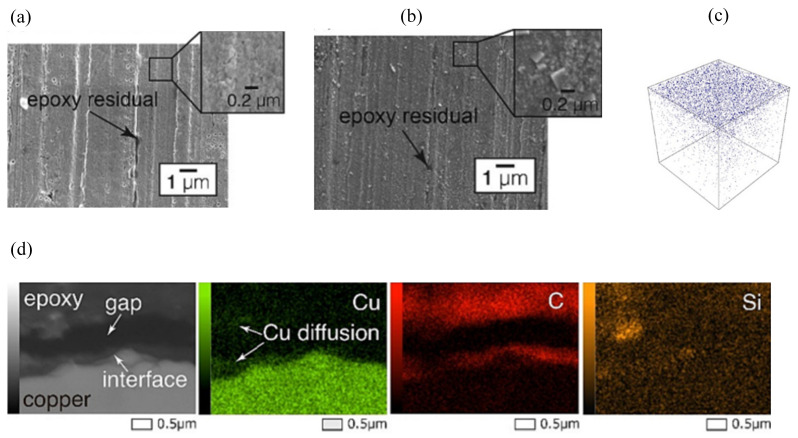
(**a**) Residual resin cleavage on the delaminated Cu side after 1000 h HST. (**b**) Residual resin cleavage on the delaminated Cu side after 96 h PCT. (**c**) Rendering of the fracture surface of delaminated copper after 96 h PCT, indicating that delamination occurs within the EMC. (**d**) Interfacial delamination and Cu diffusion observed after 96 h PCT. Reproduced with permission [53] Copyright 2023, Elsevier.

**Figure 7 micromachines-15-00376-f007:**
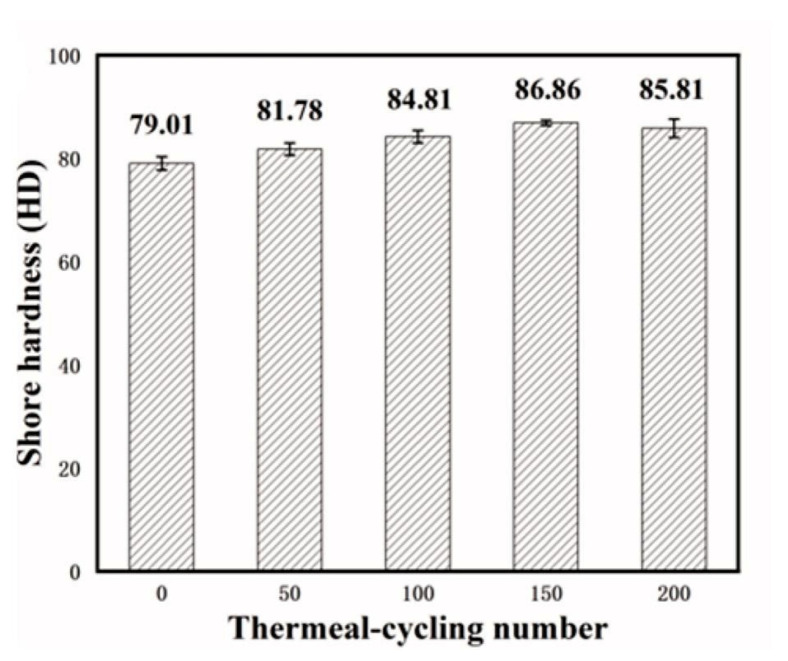
The resin undergoes oxidization and cross-linking during thermal cycling, which leads to an increase in Shore hardness and then a decrease in Shore hardness. Reproduced with permission [56]. Copyright 2022, Taylor & Francis.

**Figure 8 micromachines-15-00376-f008:**
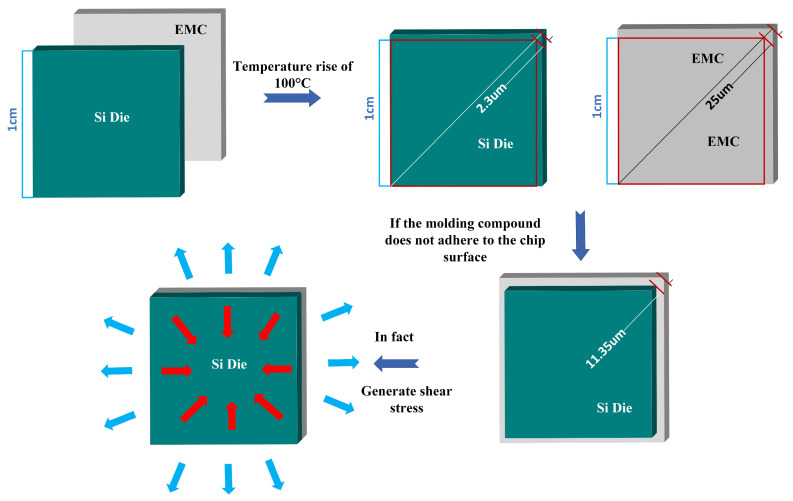
Schematic diagram of shear stress generated by CTE mismatch of Si/EMC.

**Figure 9 micromachines-15-00376-f009:**
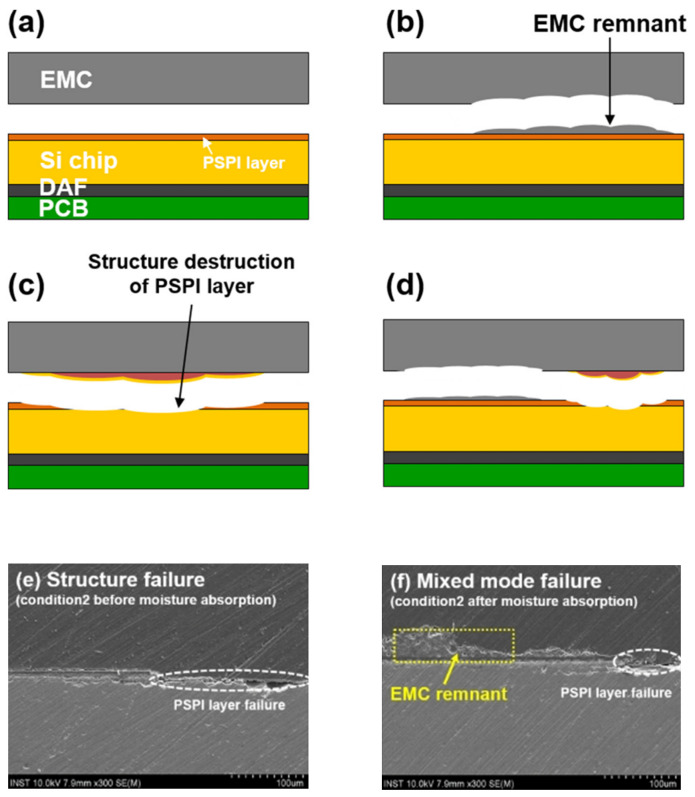
Schematics of failure mode after test: (**a**) interfacial, (**b**) cohesive, (**c**) structural, and (**d**) mixed mode failure. Cross-sectional SEM images after adhesion test: (**e**) structural failure, (**f**) mixed mode failure. Reproduced with permission [68]. Copyright 2019, Elsevier.

**Figure 10 micromachines-15-00376-f010:**
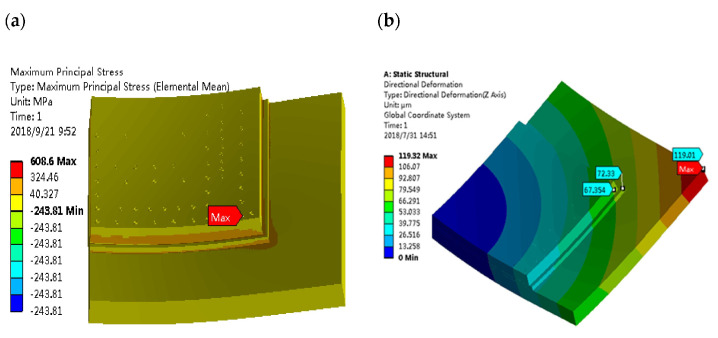
(**a**) Stress distribution in the package at 260 °C. (**b**) Warpage of the package at 260 °C. Reprinted from Ref. [21]. with open access from MDPI.

**Figure 11 micromachines-15-00376-f011:**
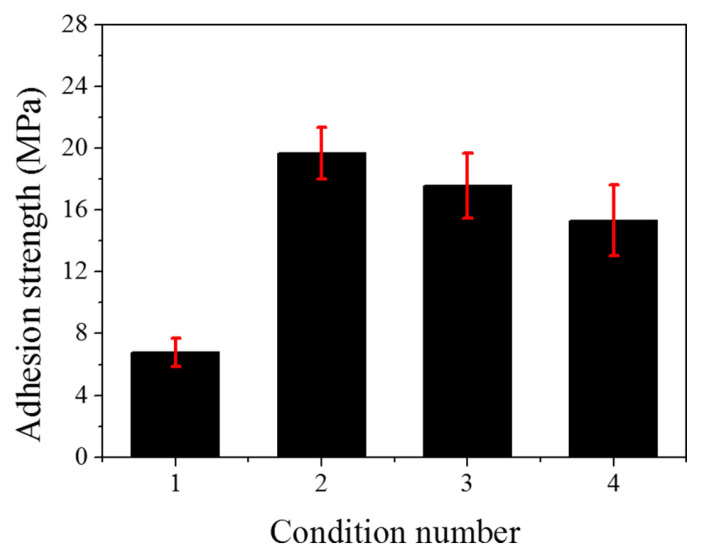
The results of an adhesion strength test for plasma treatments. Reproduced with permission. Condition 1 is the EMC/Si interfacial adhesion strength without plasma treatment; Condition 2 is the EMC/Si interfacial adhesion strength after one plasma treatment; Condition 3 is the EMC/Si interfacial adhesion strength after three plasma treatments; and Condition 4 is the EMC/Si interfacial adhesion strength after five plasma treatments [68]. Copyright 2019, Elsevier.

**Figure 12 micromachines-15-00376-f012:**
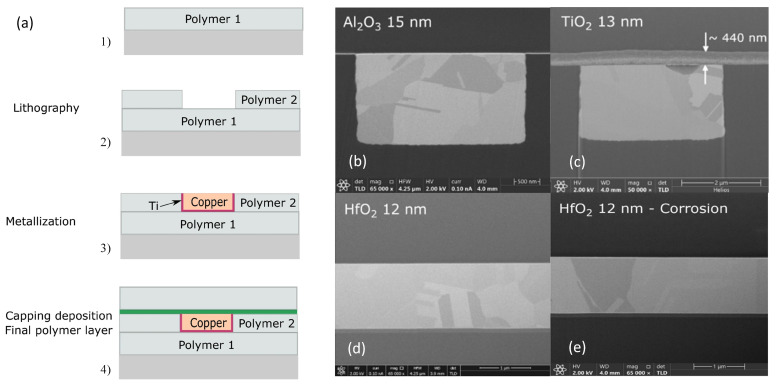
(**a**) Manufacturing process of the RDL layer with ALD technology. (**b**) RDL copper wire covered with 15 nm AI_2_O_3_ inorganic coating did not show oxidation when kept at 150 °C for 1000 h. (**c**) RDL copper wire covered with 13 nm TiO_2_ inorganic coating showed an oxidized layer of 440 nm when kept at 150 °C for 1000 h. (**d**) RDL copper wire covered with 12 nm HfO_2_ inorganic coating did not show oxidation when kept at 150 °C for 1000 h. (**e**) RDL copper wire covered with 12 nm HfO_2_ inorganic coating showed no oxidation during the corrosive stress test. Reproduced with permission [85]. Copyright 2023, Springer.

**Figure 13 micromachines-15-00376-f013:**
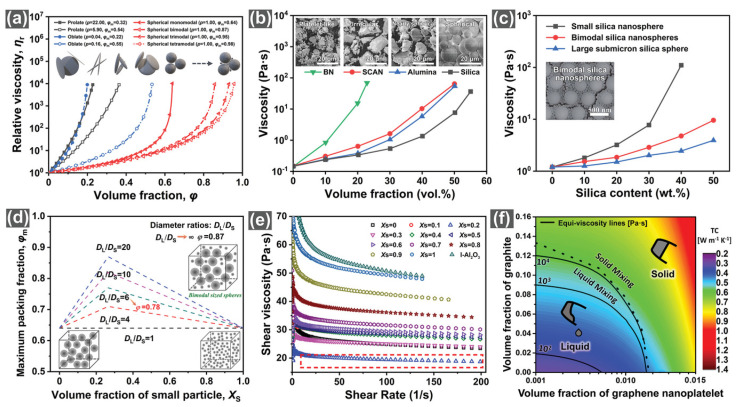
(**a**) Relative viscosity of suspensions predicted by the Krieger–Dougherty equation as a function of filler volume fraction, shape, and size distribution. (**b**) Viscosity at a temperature of 298 K and a shear rate of 5 s^−1^ as a function of filler content with different morphologies. (**c**) Viscosity of epoxy suspensions containing a small silica nanosphere (diameter: 470 ± 10 nm), a large sub-micrometer silica sphere (diameter: 50 ± 10 nm), and a bimodal silica nanosphere (58 vol% 500 ± 50 nm and 42 vol% 60 ± 10 nm) at a shear rate of 10 s^−1^. (**d**) Theoretical maximum packing of binary spherical particles and (**e**) shear viscosity of epoxy composites filled with a binary mixture of S-AI_2_O_3_ as a function of filler size distribution. I-AI_2_O_3_ stands for irregular alumina microparticles. (**d**,**e**) Reproduced with permission. (**f**) Phase diagram illustrating the effect of filler loading on viscosity and thermal conductivity of hybrid composites. Reproduced with permission [93]. Copyright 2022, John Wiley & Sons.

**Figure 14 micromachines-15-00376-f014:**
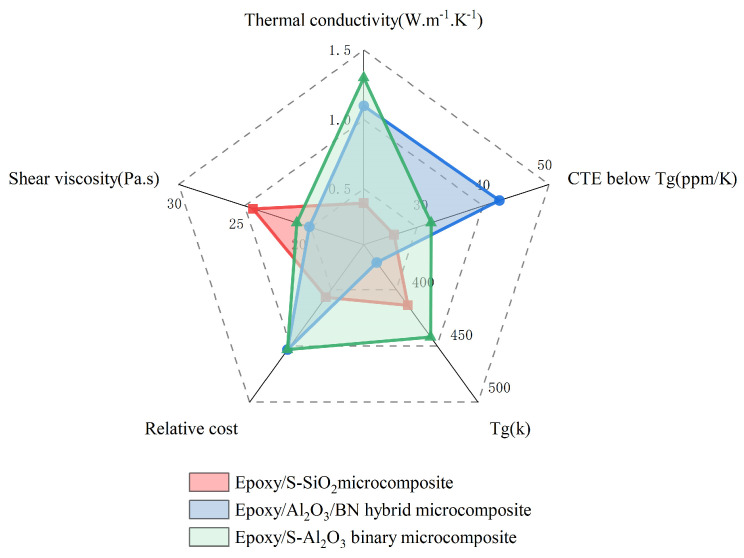
Comparison of commonly used silica-filled underfill materials: epoxy/S-SiO_2_ microcomposite; epoxy/Al_2_O_3_/BN hybrid microcomposite; epoxy/S-AI_2_O_3_binary microcomposite. Adapted with permission [93]. Copyright 2022, John Wiley & Sons.

**Figure 15 micromachines-15-00376-f015:**
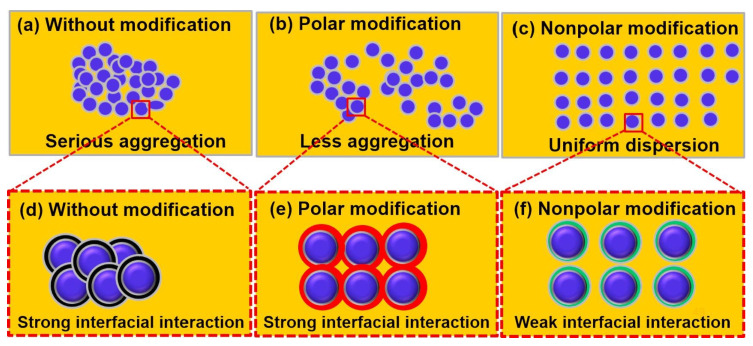
Schematic illustration of the effect of various filler modifications on (**a**–**c**) interfacial compatibility and (**d**–**f**) interfacial interaction of nanocomposite underfill. Reproduced with permission [92]. Copyright 2019, Elsevier.

**Figure 16 micromachines-15-00376-f016:**
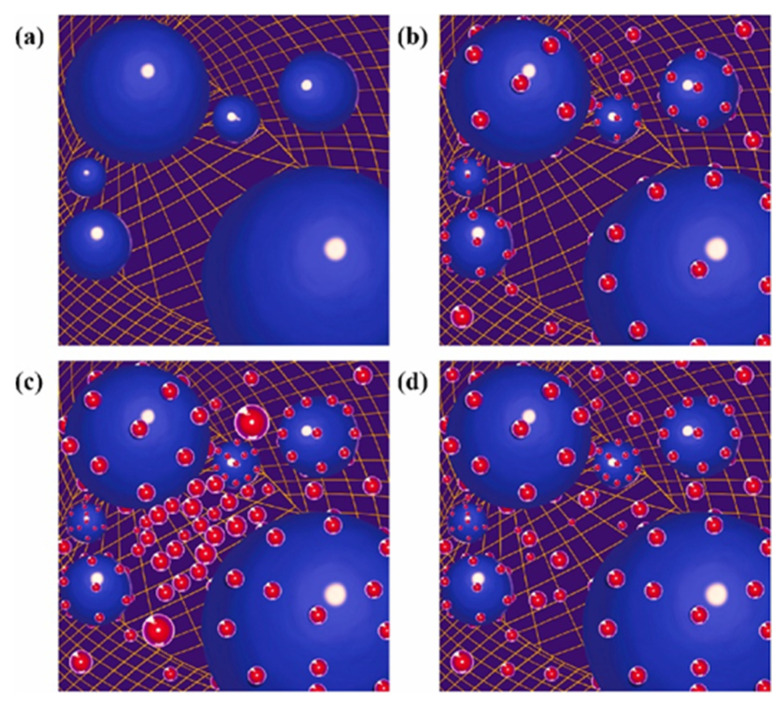
Proposed mechanism for explaining the properties of (**a**) EC, (**b**) composite with BCPs (0.5 wt%), (**c**) less-dispersed composite with BCPs (1.5 wt%), (**d**) well-dispersed composite with BCPs (1.5 wt%). Reproduced with permission [112]. Copyright 2022, Elsevier.

**Figure 17 micromachines-15-00376-f017:**
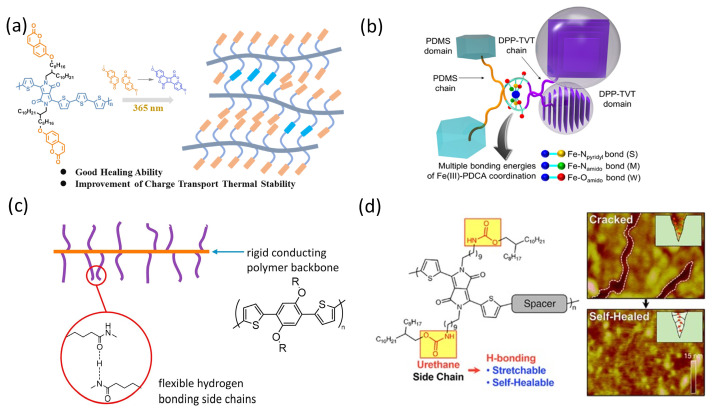
Self-healing polymer semiconductors based on hydrogen bonding, covalent bonding, and metal–ligand bonding. (**a**) Chemical structure of PDPP4T-DCM with photo-cross-linked coumarin in the side chain and schematic representation of coumarin photo-induced cross-linking of dimerized polymer chains. Reproduced with permission [116]. Copyright 2021, John Wiley & Sons. (**b**) Schematic representation of dynamic cross-linking of DPP and PDMS via Fe(III)-PDCA complexation. Reprinted from ref. [117] with open access from American Association for the Advancement of Science. (**c**) Self-healing conjugated graft copolymers based on a polythiophene phenylene (PThP) backbone and hydrogen-bonded side chains. Reproduced with permission [118]. Copyright 2018, Elsevier. (**d**) Semiconductor polymers with self-healing polyurethane side chains using hydrogen bonding for self-healing. Reprinted with permission from [119]. Copyright 2020 American Chemical Society.

**Figure 18 micromachines-15-00376-f018:**
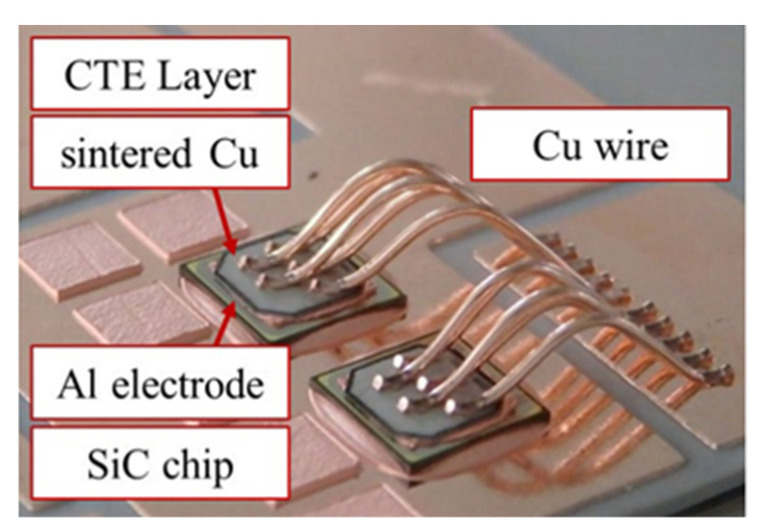
Overview of the newly developed long-lifetime chip-top packaging technology, including Cu wiring, CTE layer, sintered Cu, and SiC chip. Reprinted from ref [126]. with open access from Trans Tech Publications Ltd., Bäch, Switzerland.

**Table 2 micromachines-15-00376-t002:** Reliability test conditions. Reproduced with permission [53]. Copyright 2023, Elsevier.

Test Name	Conditions	Test Time
High-temperature storage test (HST)	200 °C	0, 240, 500, 1000 h
Temperature cycling test (TCT)	−65–150 °C dwelling time: 30 min	0, 100, 200, 500 cycles
Pressure cooker test (PCT)	121 °C, 100% RH, 2 atm	0, 96, 192 h

**Table 3 micromachines-15-00376-t003:** Temperature comparison of two flip-chip Connection Processes.

Connection Process	Chip Temperature	Substrate Temperature
Mass Reflow, MR	250 °C	250 °C
Thermo-Compression Bonding, TCB	400 °C	50 °C

**Table 4 micromachines-15-00376-t004:** Properties of ALD inorganic coating materials.

RDL Barrier Layer	Obstructing O_2_ Capabilities	H_2_O Corrosion Resistance	Low-Temperature Manufacturability
15 nm AI_2_O_3_ [84]	No oxide layer	Rapid failure	Best
12 nm HfO_2_ [85]	No oxide layer	non-corrosive	Worse
13 nm TiO_2_ [85]	440 nm oxide layer appears		Better
200 um liquid mold layers [87]	No oxide layer	non-corrosive	Best

**Table 5 micromachines-15-00376-t005:** Comparison of filler surface modification types.

	Physical Modification [104]	Chemical Anhydrous Modification [105]	Chemical Aqueous Modification [106]
Stability	Worse	Better	Best
Compatibility	Some improvement	Significant improvement	Significant improvement
Viscosity	Possible negative effects	Viscosity reduction	Viscosity reduction
Thermo-Mechanical Properties	Plasticizing effects occur	No significant improvement	Significant improvement

**Table 6 micromachines-15-00376-t006:** Comparison of the effects of different tougheners on the properties of epoxy composites.

Properties	No Toughening Agent	Carboxy End Seal [109]	2.0 wt% Silicone Rubber SR [114]	BCP M52N with 83 wt% SiO_2_ [112]
Interfacial Adhesion	Benchmark	↓	↓	M52N increased to 1.5 wt% slightly ↓
Hygroscopicity	Benchmark	↓	↑	M52N increased to 1.5 wt% slightly ↑
Delamination resistance	Benchmark	-	after 1000 cycles ↓	Continuously ↑ as M52N increases
Thermo-mechanical Properties	Benchmark	Tg ↓	Tg ↓	-
Thermal Stability	Benchmark	-	-	-
Thermal conductivity	Benchmark	-	-	↑
Toughness	Benchmark	↑	↑	Continuously ↑ as M52N increases
Mechanical Properties	Benchmark	↓	↓	Continuously ↑ as M52N increases
Fluidity	Benchmark	↓	↓	Continuously ↓ as M52N increases

Note: ↑ means increase, - means little or no change, ↓ means decrease.

## Data Availability

Not applicable.

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
