# Peer review of "Delamination of Plasticized Devices in Dynamic Service Environments"

_micromachines, 2024, doi:10.3390/mi15030376_

Round 1

Reviewer 1 Report

Comments and Suggestions for Authors

In this manuscript, the authors summarized the relevant literature and the latest research solutions on the delamination failure mechanism of polymer-based packaging devices caused by thermal stress and moisture damage in the microscopic state. The authors also provided theoretical support for subsequent research and optimization of polymer-based packages by summarizing the microscopic failure mechanism of delamination at the two-phase interface and introducing the latest solutions. However, several aspects should be improved before the article can be recommended for publication in Micromachines.

Comments:

1.       As the authors mentioned in section 2.2, thermal stress damage mainly involves three key factors: CTE mismatch, thermal cycling stress, and external mechanical stress. However, the effects of thermal stress on the two-phase interface have not been elaborated by the three factors respectively. The authors are recommended to add relevant literature and corresponding expatiations in sections to ensure the integrity of the content.

2.       The literature involved in the manuscript is merely about the mechanism of delamination failures triggered by single moisture or thermal stress damage, however, which tends to coexist in most practical applications. Hence, the authors are recommended to add some supplementary research on the cooperative delamination mechanism at the two-phase interface.

3.       In section 3.2, the research on the optimization of polymer inorganic fillers is presented. But the subsequent 3.2.2 merely introduces the single material of nanographene filler, which is not quite appropriate. the authors are recommended to delete the subsequent or add literature about other materials to make it fit the chapter theme.    

4.       There are also some minor issues to be revised. (1) Page 6 177-179, “however, when the discontinuity of the packaging material and the high temperature diffusion ambient temperature play a role, the hygrothermal analogy method is very difficult to accurately characterize the composite dynamics of the” is semantically repetitive with the next sentence. (2) Page 8 line 243, “at bottom” is repeatedly expressed. (3) Lack of literal expressions corresponding to the 4 conditions in Figure 11, which fails to underline the effects of Ar+O2 plasma treatment and may cause confusion to readers. It is recommended to check the figures and the text carefully.

Comments on the Quality of English Language

No comments

Author Response

I have revised the article according to the reviewers' suggestions:

  1. The three key factors involved in thermal stress: CTE mismatch, thermal cycling stress, and external shock have been newly read in the literature and are described and illustrated in three separate paragraphs. In addition, a separate paragraph is added to describe how external mechanical stresses (including shock and vibration) accelerate the delamination failure of devices under thermal cycling operating conditions. And some electrical interconnection structures against external mechanical stresses are added later: C4 bumps, Cu pillars, Ni/Sn pillars, and Cu/Ni/Sn pillars for enhanced support and cushioning effects.
  2. In section 2.1.2 Moisture Physical Damage several studies of delamination failure in the presence of coupled moisture-heat-electric three-phase physical fields are presented. And several new cases and mechanism summaries of delamination failure in the moisture-heat case are re-added at the 2.3 Summary.
  3. The initial inclusion of graphene nanofillers as a subsection on the same level as filler size diversification and filler surface modification was intended to emphasize the great promise of combining nanoscale and graphene in the field of fillers. Nanoscale fillers can significantly improve the performance of the overall system with only a small addition, and the change in the density of the polymer matrix is not significant. The high mechanical strength (1600 GPa), high specific surface area (2600 m2/g), and high thermal conductivity (5000 W/mK) of graphene are enough to show that it has excellent application prospects in the field of polymer fillers. Therefore, after further consideration, it was decided to retain the subsection on graphene nanofillers, and we hope that the reviewers will understand.
  4. Lines 177-179, which are semantically repetitive, have been revised; the repetition of "at bottom" in line 243 has been deleted; and the four conditions that are not clear in Fig. 11 have been explained in detail in the notes.

Reviewer 2 Report

Comments and Suggestions for Authors

The authors review a common reliability problem of non-hermetic overmolded chip assemblies: delamination happening at the interface between chip and mold epoxy under temperature cycling and high temperature  high humidity conditions. Authors provide a great overview of numerical models developed to understand failure modes, supported by analysis of empirical studies. They show that main failure contributors are thermal stresses at elevated temperatures arising due to coefficient of thermal expansion (CTE) mismatch, epoxy material volume expansion due to moisture absorption, and an influence of high vapor pressure. Also, authors investigate failures of redistributed layers (RDL) due to chemical reactions between copper and polymer resin insulators. Authors discuss several methodologies of improving package reliability: atomic layer deposition (ALD) moisture barrier introduction, optimization of inorganic and organic polymer fillers, self-healing polymer technology and layered buffer application to smoothen gradients of mechanical properties. The review is up to date based on publicly available sources. I recommend the article for publication after few minor comments are addressed.

1) Authors explain well the theoretical models to estimate moisture penetration into polymer, and they tell that experimental ways to assess moisture penetration are not precise due to local effects in assemblies. However, I think it is still worth referring to the article below that reviews some experimental methods:

X.J. Fan, S.W.R. Lee, Q. Han, Experimental investigations and model study of moisture behaviors in polymeric materials, Microelectronics Reliability, Volume 49, Issue 8, 2009, Pages 861-871, ISSN 0026-2714, https://doi.org/10.1016/j.microrel.2009.03.006.

2) In chapter 2.2 authors discuss delaminations due to thermal stress at temperature cycling test. Even though this is not exactly the topic of authors' work, it would be good to add couple lines discussing which type and size of electrical interconnects provides higher shock-absorbing properties: C4 bumps, Cu/Ni/Sn pillars, etc.

3) In the introduction to chapter 3 authors mention adhesion improvement by RDL layer plasma treatment. I think it is worth mentioning the work by Pai et al, where Si chip surface and lead frame itself are treated with plasma in order to improve adhesion and decrease delamination risks.

Pai, Hsiang Ting and Che-Hsin Lin. “Experimental and Numerical Investigation Into the Plasma Treatment and Chip Delamination in Semiconductor Packaging.” IEEE Transactions on Plasma Science 48 (2020): 3915-3920. https://doi.org/10.1109/TPS.2020.3029031

4) In chapter 3.1 authors discuss ALD layer application on copper in RDL stack. One of the limitations of this method is a low temperature budget due to polymers being stack insulators. It is also known that low temperature ALD (100-200 degrees) yields lower density films that are not ideal for moisture blocking. As a good example of ALD moisture blocking demonstration, I would mention work by Global Foundries team, where wafer-scale 300 deg ALD coating was done on photonic ICs before flip-chip process. There, moisture blocking was mostly needed to prevent change in stack refractive index, as well as establish a good interface with optical epoxy.

Z. -J. G. Wu et al., "Self-aligned Fiber Attach on Monolithic Silicon Photonic Chips: Moisture Effect and Hermetic Seal," 2023 Optical Fiber Communications Conference and Exhibition (OFC), San Diego, CA, USA, 2023, pp. 1-3, doi: 10.1364/OFC.2023.Th1A.5.

As a side note, it was very interesting to read about graphene additives and particle dispersion tuning. Very promising outlooks are envisioned.

Author Response

I have revised the article according to the reviewer's suggestions:

  1. by reading Fan's article, the difference between Fick diffusion and no-Fick diffusion of epoxy resin with increasing temperature is explained in more detail in section 2.1.1 of this paper; in addition, the two-stage diffusion model is introduced in Fan's article, and the model is described more accurately in this paper. 2. in section 2.2, an independent paragraph is added to introduce that external mechanical stresses (including shock and vibration) accelerate the delamination of devices under the operating environment of thermal cycling.
  2. In section 2.2, a separate paragraph is added to introduce that external mechanical stress (including shock and vibration) accelerates the delamination failure of the device under the operating environment of thermal cycling. And some electrical interconnection structures against external mechanical stresses are added later: C4 bumps, Cu pillars, Ni/Sn pillars, and Cu/Ni/Sn pillars for enhanced support and cushioning effects.
  3. By reading Pai's article, for plasma treatment, on the one hand, it will enhance the interfacial adhesion, on the other hand, it will also increase the moisture absorption capacity of the frame surface, and if the leadframe on the energized track will also present a high electric field gradient, which will weaken the interfacial adhesion instead. Additions are explained in the introduction section of the plasma treatment in this paper.
  4. Wu et al. applied the ALD technique to introduce a moisture barrier at 300 °C, and the fiber optic coupler showed excellent performance in overstressing and testing. Chip-level hermetic packaging was realized. The research of Wu et al. is added at the introduction of ALD technology in this paper.
